# Proteomic Markers of Aging and Longevity: A Systematic Review

**DOI:** 10.3390/ijms252312634

**Published:** 2024-11-25

**Authors:** Anna A. Kliuchnikova, Ekaterina V. Ilgisonis, Alexander I. Archakov, Elena A. Ponomarenko, Alexey A. Moskalev

**Affiliations:** 1Institute of Biomedical Chemistry, Moscow 119121, Russia; a.kliuchnikova@gmail.com (A.A.K.); alexander.archakov@ibmc.msk.ru (A.I.A.); 2463731@gmail.com (E.A.P.); 2Institute of Longevity, Petrovsky Russian Research Center for Surgery, Moscow 119435, Russia; moskalev1976@gmail.com

**Keywords:** blood plasma proteome, targeted mass spectrometry, knowledge databases, aging, longevity, high-copy proteins, biomarkers, biological age

## Abstract

This article provides a systematic review of research conducted on the proteomic composition of blood as part of a complex biological age estimation. We performed a comprehensive analysis of 17 publicly available datasets and compiled an integral list of proteins. These proteins were sorted based on their detection probability using mass spectrometry in human plasma. We propose this list as a basis for creating a panel of peptides and quantifying the content of selected proteins in the format of a proteomic aging clock. The selected proteins are especially notable for their roles in inflammatory processes and lipid metabolism. Our findings suggest, for the first time, that proteins associated with systemic disorders, including those approved by the FDA for clinical use, could serve as potential markers of aging.

## 1. Introduction

Understanding the varying rates of organ and tissue aging is crucial for pinpointing the underlying causes of age-related dysfunction that contribute to the development of diseases. The primary purpose of studying aging is to find ways to delay the onset of these diseases, prolong active life, and maintain health.

Aging occurs at different paces within different tissues and organs [1], with every individual also experiencing aging at distinct rates [2]. Furthermore, it has been observed that the relationship between aging biomarkers and age often has a nonlinear pattern [3]. Given the above, conducting a periodic comprehensive assessment throughout life is crucial. It is not only the chronological age that defines the status of aging.

According to the study of Johnson and Shokhirev [4], biological age is a complex, multifaceted concept that helps us understand why individuals age differently. While certain aspects can be quantified through various biomarkers, aging is too intricate to be fully captured by single measurements, requiring a more comprehensive approach to assess an individual’s functional aging status.

The variation in the organ aging rates in individuals [5] and the difference in models demonstrate that a particular individual may have several aging clocks specific to each tissue or organ. For example, liver age can be used to predict the course of nonalcoholic fatty liver disease, and heart disease-related mortality is a better predictor of cardiovascular age compared to liver and kidney age [6].

It is widely recognized that the primary hallmark of aging is a group of interrelated factors. Thus, these factors include genomic instability, telomere depletion, epigenetic changes, loss of proteostasis, impaired macroautophagy or regulation of nutrient uptake, mitochondrial dysfunction, stem cell depletion, altered intercellular communication, chronic inflammation and dysbacteriosis [7], as well as damage to the extracellular matrix [8] and the integrity of physiological barriers [9]. Some of these factors characterize biological age in an ambiguous manner. For example, the correlation between telomere length and chronological age is relatively low (R = 0.07), indicating the limitations of using this factor to estimate biological age [10]. It is evident that there is a clear need for mechanistic aging clocks that effectively measure damage factors, as they play a crucial role in both preventive medicine and clinical trials of geroprotective interventions.

There is a correlation between proteomic aging and significant chronic illnesses such as heart disease, liver disease, kidney disease, lung disease, diabetes, neurodegeneration, and cancer [11]. Blood is a readily accessible form of biological material that can indicate alterations in the protein profile of different tissues and organs, including the aging of single specific organs [12]. The rationale behind choosing blood plasma as the focus of this research was influenced by its well-documented nature as a study subject and the availability of mass spectrometric data and reference values for protein concentrations in healthy individuals [13,14].

Proteomic approaches offer distinct advantages for evaluating biological age compared to other molecular methods. Proteins directly reflect cellular function and phenotype, provide real-time information about ongoing biological processes, and can be measured with high precision in easily accessible biological fluids [13]. Unlike epigenetic marks or genetic variations, protein levels respond dynamically to environmental influences and therapeutic interventions, making them particularly suitable for monitoring aging progression and intervention effects.

Some progress has been made in studying the potential use of plasma protein profiles for age prediction. For example, circulating plasma protein profiles can be used to accurately predict chronological age and anthropometric indices [15]. The studies on the aging clock also involve investigating a predicted age of death [16], mortality risk [17], and longevity [18,19]. Of interest in the study of aging are the variations in this process between men and women. According to the lifespan study conducted in 2019, two-thirds of the proteins (895 out of 1379) undergoing age-related changes were found to exhibit variations depending on sex [20].

Despite the wealth of knowledge in the realm of aging studies, a critical factor in the quest for universal biomarkers of aging is their adaptability across various human populations [21]. Consequently, it is of utmost importance to verify the proposed biomarkers in multiple countries while considering variables such as gender, ethnicity, and other demographic factors.

Our work focused on identifying and compiling a list of the most universal markers among those found in several studies on different populations covering serum and plasma samples of people aged 10 to 110 years. This list is suggested as the potential basis for creating a mass spectrometric kit to quantify the representation of selected proteins in human plasma and to determine biological age. We propose to use targeting methods such as selected/multiple reaction monitoring (SRM/MRM) using synthesized isotope-labeled peptide standards to quantify potential universal biomarkers of aging. This approach enables the identification of proteins in biomaterial and provides accurate measurement of their contents, which is crucial for their medical application [13].

## 2. Materials and Methods

We selected proteins associated with aging and the development of age-related diseases that can be detected in human blood plasma by mass spectrometry. We thoroughly examined the literature to find studies that had successfully identified sets of proteins exhibiting notable changes in their representation levels throughout human aging (Figure 1).

The review was not registered; however, it was conducted in accordance with the Preferred Reporting Items for Systematic Reviews and Meta-Analyses (PRISMA) statement [22] (Appendix A, “PRISMA checklist”). The study selection process is detailed in the PRISMA flow diagram included in Figure 1. Two authors (A.A.K. and E.V.I.) independently conducted the study selection and data extraction process. In the case of a disagreement regarding the inclusion of a specific study or during the data extraction, an agreement was reached through discussion among all the authors.

The publications to be analyzed were searched for across the PubMed database using the query “proteomics aging clock” [23]. Lists of proteins found in serum or plasma and associated with aging were downloaded from the selected papers (Appendix A). References to publications in which proteins are indicated as markers of aging, as well as descriptions of the cohorts and countries in which the studies were conducted, are detailed in Appendix A.

Based on the downloaded data lists, a comprehensive list of proteins was created by specifying the number of experiments found in the proteomic databases PRIDE [24] and SRMAtlas [25]. These experiments were conducted across various human tissues and biological fluids; they also included data on the quantitative protein content in the blood plasma of a healthy individual [13]. The frequency of occurrence in datasets associated with aging and age-related diseases was calculated for each protein (Appendix A). Proteins were ranked based on frequency of occurrence and additional parameters, such as the probability of mass spectrometric detection in plasma from proteomic repository data. Additionally, the selected proteins were characterized using a list of proteotypic peptides (SRMAtlas [25]) to assess their suitability for use in a panel for biological age estimation by Selective Multiple Reaction Monitoring (SRM) (Appendix A).

The resulting list of protein identifiers was mapped to human metabolic pathways using the KEGG Mapper tool [26] available at [27] to generate biological annotations. The proteins were subjected to further characterization in terms of GeneOntology (Released 11 March 2024) using the PANTHER Overrepresentation Test (Released 20 June 2024) [28].

In addition to KEGG pathways, future analyses could benefit from incorporating other protein-specific databases, such as Reactome [29], which offers detailed pathway information specifically curated for proteins and their interactions. This would provide complementary insights into the functional relationships between aging-associated proteins.

The feasibility of protein detection by mass spectrometry in human plasma was assessed by comparing the resulting list with the list of 4390 proteins detected by the Human Plasma Proteome Project [30,31]. The data were downloaded directly from the project page [32], (Appendix A). The proteins least specific to aging development were identified by comparing the resulting list of identifiers with the most frequently detected proteins from Petrak et al. [33]. The original article provides the names of protein groups (e.g., keratins, actins, etc.). A list of protein identifiers belonging to each group was compiled manually using the built-in UniProt search engine. The clinical significance of proteins was assessed by comparing them to the list of proteins approved by the FDA for clinical trials [34] (Appendix A).

## 3. Results and Discussion

### 3.1. Proteomic Aging Clock

Olink, SOMAscan, and mass spectrometry technologies are currently the principal methods in proteomics used to explore aging. The Olink technology [35] employs pairs of DNA oligonucleotide-labeled antibodies to selectively bind the target antigen in a liquid medium. This process enables the subsequent steps of hybridization and elongation by DNA polymerase. The resulting DNA barcode fragment is then amplified using PCR. The SomaScan [36] method employs aptamers, which are DNA/RNA molecules. This platform allows native proteins in complex matrices to be quantified by converting the binding sites of individual proteins into the appropriate reagent concentration, which is subsequently measured through microarray hybridization.

According to aging research statistics available on Pubmed, out of over 6000 publications, less than 2% of studies were performed using Olink and SOMAscan technologies. Mass spectrometric approaches were used in 27% of studies, with over 160 studies analyzing blood plasma. The other studies were performed using immunoenzymatic techniques (e.g., ELISA). Before a sufficient number of experiments employing new techniques are conducted, mass spectrometric data, and SRM in particular, will be considered the optimal platform for proteomic profile assessment [13].

In our work, we selected 17 studies performed in the past decade (Appendix A). Given that aging is a lifelong process [37], we considered studies that provide information on age-associated proteins across a diverse group of individuals aged 14 to 109 years (Figure 2). Among the selected works, three studies were performed using the Olink technology [11,15,17]; five works were performed on the SOMAscan platform [16,20,38,39,40]; and six studies included mass spectrometric analysis results [18,19,41,42,43,44]. Three papers involve the data analysis of multiple sources. For example, the study by Coenen et al. provides integrated information from six studies conducted using the SOMAscan platform [45], and two studies report the results obtained using different proteomic methods [46,47]. These two papers provide the data not only from serum or plasma but also from other biological materials (liver, lateral thigh broad muscle, lacrimal fluid, saliva, dermal fibroblasts, colonic epithelial tissue, cerebrospinal fluid, urine, frontal cortex, etc.). These data were chosen to show that age-associated proteins characteristic of specific tissues can also be detected in plasma.

### 3.2. Most Frequent Proteins Associated with Aging and Age-Related Disease Development

The analysis encompassed a total of 2227 proteins, as detailed in Appendix A. A comparison of the lists obtained from the papers revealed 892 proteins common to two or more datasets. A thorough literature search for markers of aging not covered by the three main current methods revealed 17 proteins (Appendix A, column “Marker of aging”), with three of them not found by our analysis but added to the list of meta-analysis results. The first of them, CD40 ligand (P29965), one of the important markers of inflammation [48,49], is potentially detectable by mass spectrometric methods, as evidenced by the data on the number of MS experiments found in PRIDE and SRMatlas databases. Two longevity proteins, fibroblast growth factor 21 (FGF-21, Q9NSA1) and humanin (Q8IVG9), prove to be challenging to detect. For example, detecting humanin by mass spectrometric methods is not possible due to its short (24 amino acid residues) sequence unsuitable for classical proteases (trypsin or chymotrypsin) [50]. The quantitative detection of FGF-21 [51,52] at the sub-nanogram per mL level in human serum can be achieved using nanoflow liquid chromatography/tandem mass spectrometry (LC/MS/MS). However, this assay may have limited reproducibility and instability of the MS signal, limiting its use as a reliable analytical method for routine clinical applications [53].

The biological annotation of the full list of proteins (*n* = 2227) was achieved through an enrichment analysis of the obtained protein list based on Gene Ontology (GO) database categories to identify the most significant functions and processes among them. The analysis showed the proteins, according to the “Biological Processes” category of Gene Ontology (GO), were involved in regulating innate immunity, including the positive regulation of natural killer cell chemotaxis (FDR q-value = 1.78 × 10^−4^), which was found to have the primary function of binding the chemokine receptor CCR1 (FDR q-value = 3.9 × 10^−4^). In terms of localization, these proteins were identified with high certainty (FDR q-value = 9.51 × 10^−3^) as belonging to the plasma membrane signaling receptor complex.

Next, the significance of each protein’s contribution was assessed by measuring the frequency of its occurrence in datasets associated with aging and the development of age-related diseases. Twenty proteins were found to be common for eight or more datasets, with the highest number of overlaps found for pleiotrophin (P21246), macrophage metalloelastase (P39900), and natriuretic peptides B (P16860) (11 papers out of 17). Three proteins, tumor necrosis factor receptor superfamily member 6 (P25445), growth/differentiation factor 15 (Q99988), and vascular endothelial growth factor A, long-form (P15692), were found to be common to the ten datasets. The results are presented in Appendix A.

The proteins found in numerous datasets proved to be involved in a multitude of functions and have links to major chronic diseases. For example, the expression of pleiotrophin (PTN, P21246), one of the main candidates for aging markers, decreases with age in neural stem and progenitor cells. Mice lacking PTN exhibit impaired neurogenesis accompanied by poor memory and reduced learning ability. The increased expression of PTN in aging mice was found to restore neurogenesis and eliminate related memory problems [54]. In model organisms, flies carrying mutations in pleiotrophin homologous genes were shown to live shorter lives, with increased expression of these genes extending lifespan [55].

Macrophage metalloelastase (MMP12, P39900), represented in 11 of 17 datasets, is responsible for regulating adipose tissue growth and insulin sensitivity and can mediate nitric oxide production during inflammation [56]. A genetic variant at the MMP12 locus was demonstrated to be associated with atherosclerosis of large arteries [57].

Natriuretic peptide B (P16860) is increasingly recognized as a prognostic marker in patients with acute coronary syndrome [58]. Natriuretic peptides have been reported to provide a promising mechanism of action in the treatment of congestive heart failure based on their vasodilatory and diuretic properties, as well as effective inhibition of the renin-angiotensin-aldosterone system [59].

The aging process is often accompanied by modifications in neuromuscular, immunologic, and endocrine functions, which can contribute to diminished muscle and bone mass as well as frailty in older adults [60]. Tumor necrosis factor receptor superfamily member 6 (FAS, P25445) has been found to mediate TNFα-induced skeletal muscle atrophy in mice during aging [61]. Mice lacking the Fas gene experience a reduced lifespan and suffer from severe autoimmune and allergic inflammation [62]. Growth differentiation factor 15 (GDF15, Q99988), considered by Hong et al. as a modulator of bone and muscle metabolism [63], is associated with aging in 10 datasets out of 17. Additionally, experimental studies involving transgenic mice have demonstrated that increased expression of human GDF15 leads to extended lifespan and enhanced insulin sensitivity [64].

### 3.3. Selection of a Panel of Proteins for Quantitative MS Analysis

The aging-related proteins selected in this study (*n* = 2227) were mapped to several lists of proteins. The first list (1) includes proteins detected within a significant international initiative to create a catalog of human plasma proteins (Human Plasma Proteome Project, Appendix A) [30,65]. In Figure 3, this list is labeled as “HPPP” (*n* = 4390). This list represents the proteins that are not intracellular but are found in the plasma. The second list (2) presents the most frequently occurring proteins described as biological markers of various pathologies studied by proteomic methods [33]. This list is labeled as “Déjà vu” (*n* = 582) in Appendix A. The third (3) is a list of proteins approved by the FDA for use as biomarkers. This list is labeled as “FDA-approved” (*n* = 111) in Appendix A.

We have found that about 40% of proteins associated with aging are not present in any of the datasets analyzed. Most of these proteins can be detected by mass spectrometric methods, but the possibility of their detection in plasma has not been confirmed (Figure 3b). This list most likely contains proteins associated with specific signs of aging rather than systemic markers of age-related diseases. Notably, the list of proteins approved by the FDA in diagnostics is almost entirely included in the list of proteins associated with aging. This finding may be due to the fact that indicators of systemic abnormalities in molecular cascades that also lead to increased biological age have been proposed as biomarkers. Only 35% of proteins (*n* = 1493) forming the proteomic map of human blood plasma were found to be associated with aging. This rate may be attributed to blood plasma not being the primary biological material for assessing biological age.

The high frequency of proteins identified in proteomic experiments as differentially expressed in diseases may be due to the high representation of these proteins since the analytic methods record the same proteins all the time because of their availability [66]. As a result, we observe a substantial overlap in the list of differentially expressed proteins across various diseases. This finding implies that the list consists primarily of proteins that are not specific markers for any pathology, making them more akin to negative examples. A significant overlap can be observed between the proteins listed here and the proteins associated with aging. This overlapping can be attributed to recognizing aging as a contributing factor to multiple diseases, resulting in the presence of shared marker proteins. Consequently, high-copy proteins that exhibit fluctuating levels across different pathologies could be included in a protein panel aimed at evaluating systemic disorders, such as aging and age-related diseases. Ultimately, the philosophical question arises of whether diseases are a direct outcome of aging or merely a manifestation and outcome of diseases. Therefore, it seems reasonable to use such non-specific proteins to assess the relationship between changes in protein content and biological age.

To exemplify our methodology in this study, we propose a modified technique for filtering a list of proteins associated with aging (Appendix A) and then generating a panel of candidate proteins for detection using SRM (Table 1).

The proteins associated with aging were compared with those obtained from a meta-analysis of data from quantitative SRM screening of the blood plasma of healthy people [13]. Using this list for comparison offers distinct advantages. First of all, it encompasses data on a stable characteristic of the healthy human proteome, namely the proteins present in at least 70% of the analyzed samples. Moreover, these proteins demonstrate low interindividual quantitative variability, as indicated by a CV below 40%. The findings are displayed in Appendix A. A total of 167 proteins were found to be common in both lists. The GeneOntology element enrichment analysis revealed that, in terms of localization, these proteins were primarily associated with complement component C1 (FDR q-value = 2.27 × 10^−5^). The main biological processes were found to be negative regulation of complement activation by the lectin pathway (FDR q-value = 3.98 × 10^−3^) and negative regulation of plasma lipoprotein oxidation (FDR q-value = 3.95 × 10^−3^). The main function was the binding of lipoprotein particle receptors (FDR q-value = 8.41 × 10^−7^).

Next, the proteins were sorted according to their frequency of occurrence in the datasets. According to the following criterion, proteins were selected for which the average plasma concentration in healthy individuals with small interindividual variability had been determined [13]. After filtration, the list included 16 candidates (see Table 1) found in six or more datasets and reported in 10 or more MS experiments found in the PRIDE and SRMatlas databases. Also included in the list were four proteins (Vascular endothelial growth factor A, Galectin-3, and C-X-C motif chemokine 9 and 10), for which no concentration data are available but which are of great research interest as markers of aging and have been well-studied by other proteomic methods [48,49,67]. The resulting list of candidate proteins for aging research in plasma was compared with data from the “Déjà vu in proteomics” project [33]. Only one protein (Plasminogen) was found to overlap with both our list and the Déjà vu list of frequently detected proteins in various pathologies, making it non-specific to aging. Similarly, Albumin, which could also be present in the Déjà vu list and thus labeled as non-specific in Table 1, is often not included in the final lists because of depletion during sample preparation. Nevertheless, we consider it to be one of the potential candidates for the proteomic panel for biological age estimation.

The list of 20 selected proteins was mapped using the KEGG tool to search for metabolic pathways associated with aging processes. Of all the proteins listed in the resulting table, a few stood out as being of particular interest. It is an inflammaging marker vascular endothelial growth factor A (VEGFA, P15692) [48,49] involved in multiple signaling pathways, carcinogenesis, atherosclerosis and rheumatoid arthritis. Three proteins, coagulation factor X (P00742), mannan-binding lectin serine protease 1 (P48740), and complement component C9 (P02748), were involved in complement and clotting cascades. The chemokines CXCL9 (Q07325) and CXCL10 (P02778) were both found to be involved in interactions with viral proteins and were associated with the Toll-like receptor signaling pathway. Another protein of interest on the list was fibronectin, as the accumulation of damage in extracellular matrix proteins is a significant hallmark of aging [8]. We suggest that for a more comprehensive characterization of changes, attention should be paid to selecting those proteins that are found in different pathways.

The suitability of each protein for a biological age estimation panel by directed multiple reaction monitoring (SRM) was assessed by characterizing a list of proteotypic peptides using SRMAtlas [25]. The results are provided in Appendix A.

It is worth noting that the final table can be sorted in a different way, allowing the researchers to choose whether to use well- or poorly-studied proteins for the panel and whether to pay attention to interindividual variability data or frequency of occurrence in the datasets.

## 4. Conclusions

A comprehensive analysis of proteomic studies on human blood plasma was conducted to compile a list of proteins that could serve as a basis for creating a mass spectrometric kit. This kit would enable quantifying specific proteins in human blood plasma and provide insights into biological age. The comparison of quantitative data from the healthy human proteome has identified age-associated proteins with limited interindividual variability among conditionally healthy individuals. Furthermore, we have confirmed their involvement in immune and inflammatory processes, as well as lipid metabolism.

Mapping the list compiled in this study to the current level of knowledge would allow one to realize different approaches to creating the diagnostic panel. One approach is to use proteins that are highly detectable in plasma, as demonstrated by numerous studies on various cohorts, which indicate an association with age-related diseases. Investigating the role of non-specific proteins found in different diseases could be beneficial for studying aging as a systemic phenomenon. The presence of these proteins might signify overall disruptions in the body that indicate the aging process. For more comprehensive coverage of the molecular cascade that may undergo modifications, it is recommended to integrate proteins from diverse metabolic pathways into the panel for quantification using mass spectrometric techniques. Additionally, the cost and complexity of the measurements should be considered lest the panel be overloaded. It is possible that future research may establish correlations between protein quantification and biological age [68].

An alternative approach to assembling the panel is to assess various mechanistic biomarkers of aging, which have not yet been incorporated into cohort studies but are linked to aging processes. These biomarkers may include extracellular matrix degradation, genetic instability, and mitochondrial dysfunction.

Compared to other molecular clocks in the literature, our proposed panel of 20 proteins represents a balanced approach. While some existing clocks utilize fewer markers (e.g., epigenetic clocks using 3-5 CpG sites) and others employ hundreds of data points (e.g., transcriptomic clocks) [1,4], our selection aims to optimize between practical feasibility and comprehensive coverage. Previous studies have shown varying prediction accuracies, with some clocks achieving correlations of 0.65–0.85 with chronological age [10]. Future validation studies will be needed to determine how our protein panel performs in comparison. The moderate number of markers in our panel may facilitate clinical implementation while maintaining sufficient biological information to capture the complexity of aging processes.

The candidate biomarkers of aging identified in this study may have universal applicability, as their presence has been confirmed in several studies on different populations in serum and plasma samples from people of both sexes of a wide age range. In the long term, the compiled list may prove helpful in creating a routine blood test that could be easily scaled up to manage the health of larger populations.

## Figures and Tables

**Figure 1 ijms-25-12634-f001:**
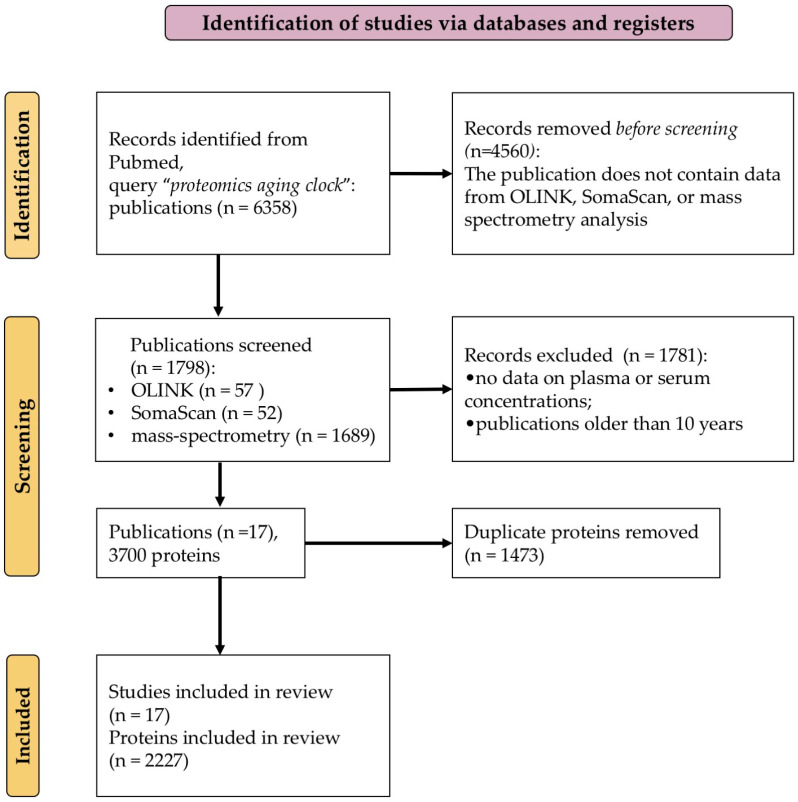
PRISMA [22] flow diagram of the literature screening and selection processes.

**Figure 2 ijms-25-12634-f002:**
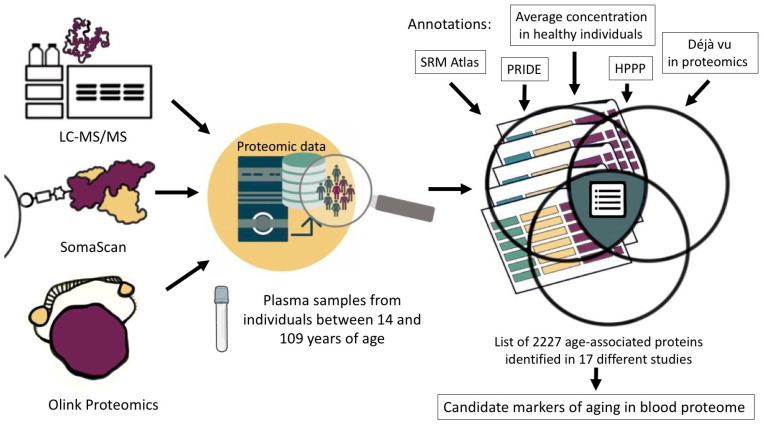
Selection of proteins associated with aging and development of age-related diseases that can be detected by mass spectrometry in human blood plasma.

**Figure 3 ijms-25-12634-f003:**
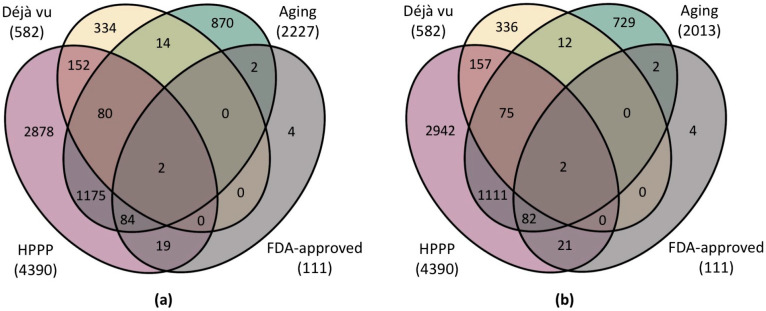
Venn diagrams showing (**a**) the intersection of the complete list of proteins associated with aging with the lists of FDA-approved proteins [34], HPPP-detected proteins [30], and the list of most frequently detected proteins (Déjà vu, [33]) (**b**) the intersection of the list of aging-associated proteins detected by mass spectrometric methods with the lists of proteins approved by the FDA [34], detected under HPPP [30], and the list of most frequently detected proteins (Déjà vu, [33]).

**Table 1 ijms-25-12634-t001:** The list of age-associated proteins suggested for the detection by SRM. Count indicates the frequency of occurrence in 17 datasets under study; MS-data represents the total number of MS experiments obtained from PRIDE and SRMatlas databases; concentration, nmol/L, indicates the average protein concentration in the plasma of a healthy individual measured by the SRM method [13]; asterisks (*) indicate the proteins that are markers of aging; proteins that are non-specific for the aging process highlighted in bold.

#	Uniprot ID	Gene Name	Protein Name	Count (Out of 17 Datasets)	MS-Data	Concentration, nmol/L
1	P15692 *	VEGFA	Vascular endothelial growth factor A	10	598	n/a
2	P01034	CST3	Cystatin-C	9	334	34.7
3	P14151	SELL	L-selectin	8	878	76.0
4	P01011	SERPINA3	α1-Antichymotrypsin	8	714	5270.0
5	P02765	AHSG	Alpha-2-HS-glycoprotein	8	544	1982.7
6	P18065	IGFBP2	Insulin-like growth factor-binding protein 2	7	553	8.5
7	P02751	FINC	Fibronectin	7	395	1252.0
8	P00742	F10	Coagulation factor X	7	330	170.3
9	P02778 *	CXCL10	C-X-C motif chemokine 10	7	239	n/a
10	P08697	A2AP	Alpha-2-antiplasmin	7	68	875.4
11	P01031	CO5	Complement C5	6	702	150.8
12	P02679	FIBG	Fibrinogen gamma chain	6	588	5384.0
**13**	**P02768**	**ALB**	**Albumin, serum**	**6**	**583**	**1,101,000.0**
14	P17931 *	LGALS3	Galectin-3	6	408	n/a
15	P02748	C9	Complement component C9	6	368	259.0
16	O14791	APOL1	Apolipoprotein L1	6	355	692.5
17	P05155	IC1	Plasma protease C1 inhibitor	6	344	399.5
**18**	**P00747**	**PLMN**	**Plasminogen**	**6**	**267**	**661.5**
19	P48740	MASP1	Mannan-binding lectin serine protease 1	6	123	89.9
20	Q07325 *	CXCL9	C-X-C motif chemokine 9	6	12	n/a

## Data Availability

All authors confirm that all data and materials support their published claims and comply with field standards. These are included in this article and Appendix A.

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
