# Peer review of "Proteomic Markers of Aging and Longevity: A Systematic Review"

_ijms, 2024, doi:10.3390/ijms252312634_

Round 1
Reviewer 1 Report
Comments and Suggestions for Authors
This is a thoughtful and timely review of proteomic aging biomarkers. I only have three minor comments.
1) Abstract: In the first sentence, the authors write "proteomic composition of blood in relation to the assessment of biological age". Biological aging is too complex to quantify with a single metric. Just like it wouldn't be reasonable to say that grip strength or VO2 max measure biological age, it's not reasonable to conflate a protein or proteomic aging clock with full-body biological aging. As such, I would recommend using more cautious language so as not to confuse non-experts.
2) Introduction: As I describe in more detail in my recent paper (PMID: 39392224), I have to disagree with the authors that biological age is "a quantitative indicator of the individual aging rate". Instead, biological age is an abstract, qualitative concept that helps us understand why people age differently and summarizes a person's functional aging. Aging is too complex to be fully captured by a single biomarker.
3) Results and Discussion: Is there a reason the authors just looked at Gene Ontology and not at other databases well-suited for proteins, such as Reactome?
Author Response
We thank the reviewer for the time and effort spent revising the manuscript. We have adjusted the text in accordance with the comments provided, and in our opinion, this has significantly improved the article.
Comment 1: We agree that the expression “the proteomic composition of blood in relation to the assessment of biological age” is too ambitious and not entirely correct in this context. In the revised version of the article, we replaced this sentence with “the proteomic composition of blood as part of a complex biological age estimation”.
Comment 2: Suggested edit in Introduction (paragraph 3): “According to the study of Johnson and Shokhirev [4], biological age is a complex, multifaceted concept that helps us understand why individuals age differently. While certain aspects can be quantified through various biomarkers, aging is too intricate to be fully captured by single measurements, requiring a more comprehensive approach to assess an individual's functional aging status”.
Comment 3: Suggested edit in Materials and Methods (after KEGG Mapper section):
Add: "In addition to KEGG pathways, future analyses could benefit from incorporating other protein-specific databases such as Reactome [29], which offers detailed pathway information specifically curated for proteins and their interactions. This would provide complementary insights into the functional relationships between aging-associated proteins."
Reviewer 2 Report
Comments and Suggestions for Authors
The manuscript by Kliuchnikova et al. presents of a review of the current literature to identify a list of proteins that could potentially be quantified as a means of determining human biological age. The authors screened through thousands of potentially relevant studies for those using Olink, SomaScan, or mass spectrometry to quantify proteins from blood of people of varying ages. The authors eventually chose to focus on 17 relevant studies, and identified an initial list of over 2,000 candidate proteins. Further comparisons to other datasets, including proteins measured by selected reaction monitoring and found to be reproducibly detected from different individuals with low interindividual variability, resulted in a final list of 20 candidate proteins. The review is carried out well, and the authors thoughtfully consider criteria for including proteins in their final list. They provide a substantial amount of information from their review of candidate proteins in Supplemental Table 1 that other researchers may find useful. I think this review will be of interest to many readers interested in aging, and also provides a practical suggestion for monitoring biological age that could be further developed by other groups. I think there should be some improvements to the presentation and clarifications in parts of the manuscript prior to publication, though, and I describe those in more detail below.
1. In the Introduction, it would be good to briefly discuss advantages of using proteomic data or protein levels to evaluate biological age, as opposed to other types of molecular data, such as changes in epigenetic marks. This would not have to be a long section, but it would be helpful to give readers an impression of specific advantages of quantifying proteins for measuring biological age.
2. Lines 60-66 in the Introduction call attention to the need to have universal markers that would be meaningful regardless of gender and ethnicity. The authors mention the country in which the 17 studies were conducted in Supplemental Table 1a, but there is no indication about gender representation in the studies. Considering the emphasis in the Introduction on possible sex-specific differences, it would be worthwhile to add information about the proportion of samples from different genders in Table 1a. If the authors do not have that information from all the studies, it would be good to at least acknowledge that. Also, the final list of 20 candidate proteins includes candidates from six or more datasets (line 283). There are multiple datasets from the same country in the list of 17 studies, so it would be good for the authors to provide some information about whether those 20 proteins are common to datasets from most of the six countries in which studies were conducted (China, Iceland, Spain, Sweden, UK, USA), or if many of them are in datasets from just a couple of the countries. This would be relevant for how universal the candidate proteins may be.
3. Section 3.1: I think that this section would read better if the information was presented in a different order. The authors start with the 17 studies included, but then later discuss the relevant technologies and how many of the total publications analyzed utilized particular technologies. It seems that it would be easier for readers to follow if the authors first introduce the different technologies by including very brief descriptions of Olink and SomaScan (such as one sentence describing each). Figure 2 provides good graphics for these technologies, but they are not described in the figure legend or text. Considering that <2% of the 6,000 studies used these technologies, readers may not all be familiar with how they work. If the authors include the information from lines 151-157 next, then they could finish with describing the 17 studies that they move forward with for their subsequent analyses.
4. In lines 159-161, the authors refer to a set of 2,227 proteins and then 892 proteins from two or more datasets. In the next paragraph, line 178, the authors refer to “all the proteins” having Gene Ontology related to innate immunity. The authors should clarify what number of proteins or set of proteins that they are referring to by “all the proteins,” because it would be surprising to see that several hundred or more proteins were all involved in innate immunity. This also applies to the later more specific statements about NK cell chemotaxis, binding CCR1, and localization to the plasma membrane signaling receptor complex. If the authors simply mean that those are the most significant functions/processes identified amongst the proteins, that should be clarified, so that readers do not think all of the hundreds of proteins have these functions.
5. Section 3.3: It would be helpful again to start with the broadest information and data comparisons, and then work towards the final set of proteins shown in Table 1. For instance, it would read better to move lines 221-233 to be just after lines 276-278. Making that change would allow the authors to start this section by comparing the full set of over 2,000 proteins to several other relevant datasets and discussing their observations and thoughts about those comparisons (Figure 3). Then they can discuss how they used the SRM dataset in reference 12 to help reduce their list to only 20 proteins at the point when they emphasize their strategy of developing a list for detection using SRM.
6. Lines 289-291 refer to both plasminogen and albumin as being non-specific proteins, but I think it would be helpful for readers to clarify that non-specific means that they were present in the Déjà vu list. That list is mentioned in the prior sentence, but I still think some readers may not connect that to the use of the term non-specific.
7. In the Conclusion, it would be good if the authors commented on how their panel of proteins compares to other potential molecular clocks for aging in the literature. For instance, are many other clocks making use of lesser or greater numbers of data points (more or less than 20 individual measurements) and how well other clocks are predicting biological age to add more emphasis on the need for improved molecular clocks for aging. The authors cite multiple references that discuss aging clocks (such as refs 1, 4, 9, 10, 16), and could likely do this without needing additional references.
8. Lines 279-287. It seems that the authors should state that their filtered list contained 16 proteins, and then they added four more proteins they considered to be important to obtain a list of 20 proteins. This is a very minor point, but it seems that only 16 proteins met the criteria from lines 279-282.
9. Supplementary Table 1:
It would be good to make the first column of the Supplemental Information tab wide enough to fully display the column headings for the description of Supplemental Table 1b.
Line 87 in the main text refers to a “PRISMA checklist” in Supplemental Table 1, but I did not see a tab with that name in the spreadsheet. The Supplemental Information tab that gives an overview of all the tables also does not mention a checklist. It would help if the authors clarify whether there is a specific checklist table, or if the statement in line 87 just refers to the various types of information that they considered when reviewing literature and datasets.
Author Response
We appreciate the reviewer for their time and effort in revising the manuscript. We have made adjustments to the text based on the provided comments, and we believe this has greatly enhanced the quality of the article.
Comment 1: Suggested edit in Introduction (before the paragraph starting with "Some progress has been made"):
Add: "Proteomic approaches offer distinct advantages for evaluating biological age compared to other molecular methods. Proteins directly reflect cellular function and phenotype, provide real-time information about ongoing biological processes, and can be measured with high precision in easily accessible biological fluids [13]. Unlike epigenetic marks or genetic variations, protein levels respond dynamically to environmental influences and therapeutic interventions, making them particularly suitable for monitoring aging progression and intervention effects."
Comment 2: We thank the reviewer for the comment. Information about gender representation has been added to Supplemental Table 1a (column J). Additionally, the final list of 20 candidate proteins is correlated with the countries of study in which they were found. The results are presented in the updated Supplemental Table 1.
Suggested edit in Materials and Methods (before the before the paragraph starting with "Based on the downloaded data lists"): "References to publications in which proteins are indicated as markers of aging, as well as descriptions of the cohorts and countries in which the studies were conducted, are detailed in Supplementary Table 1a-1b."
Comment 3: We thank the reviewer for his valuable comment. The order of paragraphs in section 3.1 has been changed.
Comment 4: Thank you for pointing this out. Clarifying changes have been made to the text (Section 3.2, paragraph 2): “The biological annotation of the full list of proteins (n = 2,227) was achieved through an enrichment analysis of the obtained protein list based on Gene Ontology (GO) database categories to identify the most significant functions and processes among them.”
Comment 5: We agree with the comment. The text has been moved.
Comment 6 : The clarification has been made. Suggested edit:
From: "Only one non-specific protein (Plasminogen) proved to overlap with the two lists. Albumin, also labeled as non-specific in Table 1..."
To: "Only one protein (Plasminogen) was found to overlap with both our list and the Déjà vu list of frequently detected proteins in various pathologies, making it non-specific to aging. Similarly, Albumin, which could also be present in the Déjà vu list and thus labeled as non-specific in Table 1, is often not included in the final lists because of depletion during sample preparation. "
Comment 7: We agree with the comment. Suggested edit in Conclusions (before the final paragraph):
Add: "Compared to other molecular clocks in the literature, our proposed panel of 20 proteins represents a balanced approach. While some existing clocks utilize fewer markers (e.g., epigenetic clocks using 3-5 CpG sites) and others employ hundreds of data points (e.g., transcriptomic clocks) [1, 4], our selection aims to optimize between practical feasibility and comprehensive coverage. Previous studies have shown varying prediction accuracies, with some clocks achieving correlations of 0.65-0.85 with chronological age [10]. Future validation studies will be needed to determine how our protein panel performs in comparison. The moderate number of markers in our panel may facilitate clinical implementation while maintaining sufficient biological information to capture the complexity of aging processes."
Comment 8: Thank you very much for your comment, the text has been corrected.
Comment 9: We thank the reviewer for the comment. All changes were made in the Supplemental Table.